# Anatomical Prior Guided Spatial Contrastive Learning for Few-Shot Medical Image Segmentation

Wendong Huang
Chongqing Key Laboratory of Image Cognition, Key Laboratory of Cyberspace Big Data Intelligent Security, Ministry of Education, Chongqing University of Posts and Telecommunications
Chongqing, China
D220201013@stu.cqupt.edu.cn

Jinwu Hu
Pazhou Lab, School of Software Engineering, South China University of Technology
Guangzhou, China
202310189376@mail.scut.edu.cn

Xiuli Bi
Chongqing Key Laboratory of Image Cognition, Key Laboratory of Cyberspace Big Data Intelligent Security, Ministry of Education, Chongqing University of Posts and Telecommunications
Chongqing, China
bixl@cqupt.edu.cn

Bin Xiao*
Chongqing Key Laboratory of Image Cognition, Key Laboratory of Cyberspace Big Data Intelligent Security, Ministry of Education, Chongqing University of Posts and Telecommunications
Chongqing, China
xiaobin@cqupt.edu.cn

## ABSTRACT

Few-shot semantic segmentation has considerable potential for low-data scenarios, especially for medical images that require expert-level dense annotations. Existing few-shot medical image segmentation methods strive to deal with the task by means of prototype learning. However, this scheme relies on support prototypes to guide the segmentation of query images, ignoring the rich anatomical prior knowledge in medical images, which hinders effective feature enhancement for medical images. In this paper, we propose an anatomical prior guided spatial contrastive learning, called APSCL, which exploits anatomical prior knowledge derived from medical images to construct contrastive learning from a spatial perspective for few-shot medical image segmentation. The new framework forces the model to learn the features in line with the embedded anatomical representations. Besides, to fully exploit the guidance information of the support samples, we design a mutual guidance decoder to predict the label of each pixel in the query image. Furthermore, our APSCL can be trained end-to-end in the form of episodic training. Comprehensive experiments on three challenging medical image datasets, *i.e.*, CHAOS-T2, MS-CMRSeg, and Synapse, prove that our method significantly surpasses state-of-the-art few-shot medical segmentation methods, with a mean improvement of 3.61%, 2.30%, and 6.38% on the Dice score, respectively.

*Corresponding author.

## CCS CONCEPTS

• **Computing methodologies → Image segmentation**.

## KEYWORDS

Few-Shot Segmentation, Medical Image Segmentation, Contrastive Learning

**ACM Reference Format:**
Wendong Huang, Jinwu Hu, Xiuli Bi, and Bin Xiao. 2024. Anatomical Prior Guided Spatial Contrastive Learning for Few-Shot Medical Image Segmentation. In *Proceedings of the 32nd ACM International Conference on Multimedia (MM '24), October 28-November 1, 2024, Melbourne, VIC, Australia* ACM, New York, NY, USA, 10 pages. https://doi.org/10.1145/3664647.3680558

## 1 INTRODUCTION

Automatic semantic segmentation on medical images is a vital prerequisite for various clinical trials and medical image analysis, such as treatment planning [1, 26], radiation therapy [16, 21], and disease diagnosis [11, 15]. When massive labeled data is available, segmentation models can achieve impressive results. However, in many clinical practices, segmentation models are typically susceptible to a severe lack of labeled data due to rare abnormal organ examples and the high cost of expert-level dense annotations.

Recently, few-shot learning (FSL) [32] is presented as a promising solution to effectively address the above challenges, and few-shot learning based semantic segmentation methods are typically referred to as few-shot semantic segmentation (FSS) [24]. The key idea of FSS aims to employ class information extracted from labeled samples (called *support*) to instruct the segmentation of unlabeled images (called *query*). Accordingly, FSS typically learns transferable knowledge on multiple few-shot tasks (*episodes*) from *base* classes in the form of episodic learning, and episodically segment *novel* classes during inference.

FSS methods have made significant advancements in the past few years, but the majority of them focus on segmentation tasks on natural images [2, 13, 14, 24, 33, 38], while FSS methods tailored

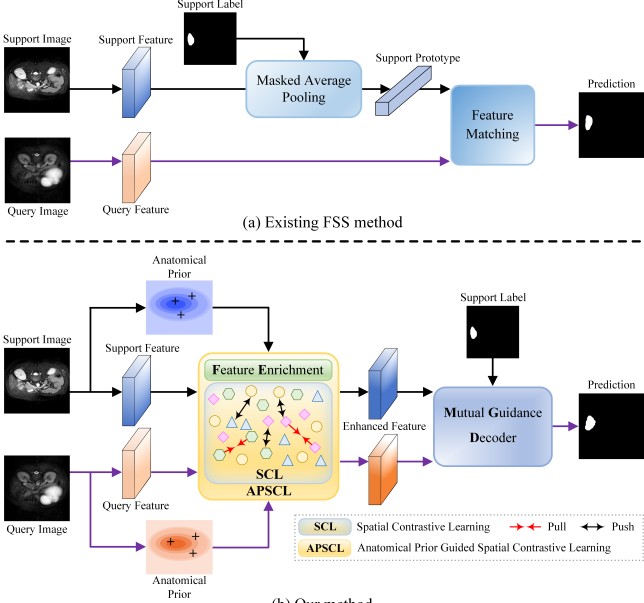

**Figure 1: Comparison between (a) existing FSS methods and (b) our APSCL. (a) Existing FSS methods only depend on support prototypes to guide the segmentation of the query image, which hinders the discriminability of medical image features. (b) Our APSCL not only takes into account the segmentation guidance of support prototypes for the query image, but also fully exploits anatomical prior knowledge contained in both the support image and the query image for feature enhancement, which is conducive to better base class separation and novel class generalization.**

for medical images are quite rare [10, 23, 35]. This may be due to the difference between the intrinsic background of natural and medical images. Current FSS methods on natural images try to prevent the class representation from being mixed with semantic information from the background features, because complex and diverse background information tends to interfere with the segmentation results of the model. Differently, the distribution of organs in medical images (such as MRI or CT) is relatively stable. Other organs in the background features tend to contribute to locating and scoping the target organ. These background features should also be fully utilized for few-shot medical image segmentation, referred to as anatomical prior knowledge. Hence, FSS methods for natural images cannot be directly adopted to solve medical image segmentation tasks in few-shot scenarios. In the past few years, some works try to introduce FSL to solve segmentation problems of medical images in few-shot settings. Roy *et al*. [23] design a prototypical learning framework based on squeeze and excitation blocks for multi-organ segmentation in CT scans via FSL. Kim *et al*. [10] devise a bidirectional recurrent neural network based few-shot segmentation framework for volumetric medical image segmentation. Huang *et al*. [6] generate adaptive prototype vectors from a vector quantization perspective and perform dense prediction by cosine similarity. Nevertheless, as depicted in Fig. 1 (a), the essential anatomical prior knowledge in medical images has not been

fully considered and integrated into the existing FSS methods to solve the medical image segmentation problem in low-data regimes, which may lead to degradation of segmentation performance.

To solve the aforementioned issue, we propose a novel anatomical prior guided spatial contrastive learning for medical image segmentation in few-shot scenarios, which fully exploits the limited annotations to improve the discriminability of feature representation, as depicted in Fig. 1 (b). We refer to the proposed method as APSCL. To be specific, we incorporate anatomical information obtained from the prior distribution into the embedding features and then employ these features to construct spatial contrastive learning, to improve the distinguishability of the features learned by the model in data-scarce clinical medical scenarios. Furthermore, a novel mutual guidance decoder is proposed to leverage the guidance information in support samples to activate the co-occurrent features between support and query branches in an interactive guidance manner, which is beneficial to yield better segmentation results. Meanwhile, the spatial contrastive learning model combined with the new mutual guidance decoder enables our framework to be effectively trained end-to-end by the episodic learning paradigm for few-shot medical image segmentation. We evaluate our framework on three publicly available medical image segmentation datasets, *i.e.*, CHAOS-T2 [9], MS-CMRSeg [42], and Synapse [12], in the 1-way 1-shot segmentation task.

To recap, our contributions are as follows:

- We propose to fully exploit anatomical prior knowledge inherent in both the support image and the query image to construct contrastive learning, from a spatial perspective, to facilitate the model to learn more distinguishable embedding features. To the best of our knowledge, this is the first time that anatomical prior knowledge is explicitly leveraged to solve a medical FSS problem via a probabilistic network.
- We propose a mutual guidance decoder to yield the prediction mask for the query image, which can effectively recognize the query features derived by the model combined with spatial contrastive learning.
- The proposed method significantly surpasses current state-of-the-art FSS methods and sets the new state-of-the-art FSS performance on three public medical image datasets.

## 2 RELATED WORK

### 2.1 Few-Shot Semantic Segmentation

Few-shot learning (FSL) aims to learn transferable knowledge from limited annotated samples, which has been extensively used for image classification [32, 40], object detection [3, 8], and semantic segmentation tasks [2, 24]. For segmentation tasks, few-shot segmentation methods aim to achieve dense pixel prediction of unseen classes under the guidance of a small number of annotated samples. Shaban *et al*. [24] first introduce FSL to solve segmentation tasks in low-data regimes and design a two-branch segmentation framework. Dong *et al*. [2] design an episode-based prototypical segmentation network, which yields a representative prototype vector for each semantic category from few support samples and then uses these prototypes to instruct the segmentation of the query image. The prototypical paradigm is widely adopted in later research works [13, 33] and gradually applied to medical image segmentation

tasks with extremely limited labeled data [7, 10, 23]. Roy *et al.* [23] first introduce prototypical learning with squeeze and excitation mechanisms to solve the abdominal organ segmentation problem in volumetric CT data. Wu *et al.* [35] employ semantic information from non-target slices to construct a multi-view contrastive learning strategy for medical image segmentation under few-shot scenarios.

The recent studies mentioned above utilize support data to compute class prototypes and then use them to guide the segmentation of query images, which does not fully exploit the potential of the limited annotated data during training. In contrast, we propose to leverage anatomical prior knowledge in finite labeled data to construct contrastive learning, which contributes to boosting the discriminability of the learned features of the model.

## 2.2 Probabilistic Distribution Learning

The latent diffusion model (LDM) [22] is a probabilistic model that learns a data distribution by progressively denoising a variable derived from a normal distribution, which is equivalent to learning the inverse procedure of a specific Markov chain. Constraining the variance between the approximate probabilistic distribution and the normal distribution is often regarded as a regularization technique, which is used for image segmentation tasks in cross-modal scenarios [36, 37, 39]. In this way, the network can learn a shared latent embedding space with the domain-invariant characteristic from input data through regularization. In this work, we further employ regularization to constrain the distribution difference between the latent embeddings of the input sample and the corresponding label, so that the network captures the intrinsic anatomical prior knowledge of medical images and facilitates the construction of contrastive learning.

## 2.3 Contrastive Learning

Contrastive learning is aimed at learning low-dimensional embedding representations of input samples by optimizing contrastive losses, which estimate the distances of various samples in the feature space. In the past few years, methods based on contrastive learning have proliferated and obtained state-of-the-art performance in common computer vision tasks. Accordingly, contrastive learning is gradually explored to solve few-shot learning problems [17, 18]. InfoPatch [17] improves contrastive learning by employing various data augmentations and the patch-wise relationship. Ouali *et al.* [18] devise a spatial contrastive loss to learn local discriminative features to help few-shot classification. In contrast, there are currently only a few contrastive learning based methods for FSS tasks. Wu *et al.* [35] construct dual contrastive learning to solve few-shot medical image segmentation tasks by exploiting information in non-target slices. By combining intrinsic properties of medical images with contrastive learning, we devise an anatomical prior guided spatial contrastive learning to solve the problem of medical image segmentation in the few-shot setting.

## 3 METHODOLOGY

In this section, we first provide a detailed problem formulation for few-shot semantic segmentation (FSS). We then detail the proposed anatomical prior guided spatial contrastive learning (APSCL)

and mutual guidance decoder (MGD). Finally, we present a novel training objective that can be optimized end-to-end.

## 3.1 Problem Formulation

Few-shot semantic segmentation aims to train a model using a labeled training dataset $D_{train}$ that can generalize well to a testing dataset $D_{test}$ with only a handful of labeled images from unseen semantic classes, without retraining the model. In the FSS setting, training semantic classes $C_{train}$ in $D_{train}$ and unseen testing semantic classes $C_{test}$ in $D_{test}$ are disjoint, namely, $C_{train} \cap C_{test} = \varnothing$. Assume $N$ represents the total number of semantic classes in $C_{test}$ and $K$ represents the number of samples corresponding to each semantic class in $C_{test}$. The FSS problem is also known as the $N$-way $K$-shot segmentation sub-problem.

To quickly adapt to new classes and achieve segmentation in testing time, we adopt the episodic training for FSS, which is extensively employed in prior works [19, 35]. To simulate few-shot scenarios in the testing phase where each new class contains only $K$ labeled samples, the episodic training paradigm randomly samples a series of episodes from $D_{train}$ for meta-training. Each episode is comprised of a support set $S_{train}$ and a query set $Q_{train}$. The support set $S_{train} = \{(x_s^i, y_s^i(c_j))\}$ consists of the images $x_s^i$ and the corresponding ground-truth masks $y_s^i(c_j)$, and is regarded as the reference of the current class $c_j$ to be segmented; the query set $Q_{train} = \{x_q\}$ consists of unlabeled images to be segmented. Here, the superscript $i = 1, 2, 3, ..., K$ refers to the $i$-th image-mask pair of in $S_{train}$, and $j = 1, 2, 3, ..., N$ refers to the number index of class $c \in C_{train}$. During testing time, $D_{test}$ is designed in the same paradigm, but where testing images and labels are of unseen semantic classes $C_{test}$. Notice that the background class marked as $c_0$ is not included in $C_{train}$ and $C_{test}$.

## 3.2 Anatomical Prior Guided Spatial Contrastive Learning

Since medical images are inherently low in contrast visibility, the boundary between the background tissues and the target organ is relatively blurred, which makes it challenging for the FSS model to yield desirable segmentation results on target organs. Besides, in low-data regimes, there may be extreme discrepancies in morphology and scale between support and query objects from the same category, called spatial inconsistency, which further undermines the performance of FSS methods. Hence, we explore the rich anatomical prior knowledge in medical images and from the perspective of spatial information to learn discriminative features that can well distinguish the foreground target from the background tissues. Accordingly, we construct anatomical prior guided spatial contrastive learning (APSCL) with support and query features, as illustrated in Fig. 2.

**Anatomical prior generation with distribution consistency.** Inspired by the distribution regularization in LDM [22], we seek to reduce the distribution discrepancy between the latent embedding space of the input sample and the corresponding label from the perspective of regularization. Specifically, we first employ a feature encoder and a label encoder to map the input samples and labels to latent embedding variables, respectively. Afterwards, the

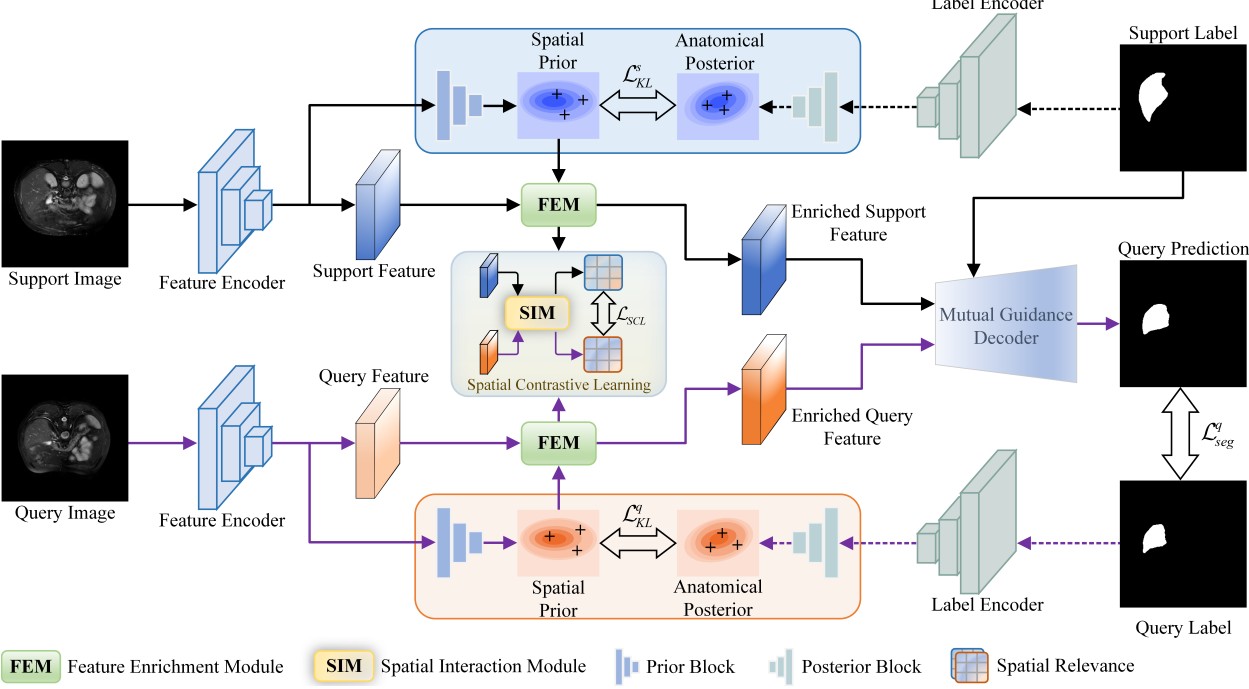

**Figure 2: Workflow of our APSCL under the 1-way 1-shot scenario. The proposed method mainly consists of feature enrichment module (FEM), spatial contrastive learning (SCL), and mutual guidance decoder (MGD). The FEM is first proposed to enrich support and query features extracted by a CNN backbone with anatomical prior knowledge learned using distribution consistency. Then, the SCL is designed to exploit enriched features to construct contrastive learning from a spatial perspective, which acts as an auxiliary branch to facilitate the segmentation task. Finally, the MGD is introduced to interactively leverage the guidance information of support samples to yield the query segmentation result.**

approximate posterior distribution can be inferred in a parametric variational manner. The anatomical structure knowledge to be segmented, referred to as anatomical prior, can be captured by regularizing the discrepancy between the distribution of the input sample and its label.

In our proposed network, a feature encoder $f_\theta(\cdot)$ is employed to embed the input samples $x$ into the feature space. The embedding representation is composed of a set of feature maps that are rich in abundant spatial information (*i.e.*, spatial prior) of input sample $x$ in multiple channels. Given an input sample $x$, the corresponding embedding representation in the embedding space $\Lambda$ is represented as $f_\theta(x)$. Since the latent embedding space $\Omega$ of the annotated ground-truth mask is mainly composed of the anatomical structure knowledge (*i.e.*, anatomical posterior) of organs, we express the anatomical embedding representation of ground-truth mask $y$ as $f_\phi(y)$, where $f_\phi(\cdot)$ is a label encoder. Afterwards, $f_\theta(x)$ and $f_\phi(y)$ from the feature and label encoders are further fed into prior and posterior blocks with one flattening layer and three sequential linear layers, where these embedding representations are exploited to initialize the probability distributions of the latent spaces. In the latent representation space, the latent factor corresponding to the input sample is denoted as $z_x$, and the latent factor corresponding to the ground-truth mask is denoted as $z_y$. The approximate posterior probability distributions of $z_x$ and $z_y$ are

denoted as $P(z_x|x,\Lambda)$ and $P(z_y|y,\Omega)$, separately. In this way, the discrepancy between $P(z_x|x,\Lambda)$ and $P(z_y|y,\Omega)$ can be considered as an effective regularization term for spatial contrastive learning and few-shot segmentation, which is formulated as:

$$\mathcal{L}_{KL} = \mathcal{D}_{KL}(P(z_x|x,\Lambda)||P(z_y|y,\Omega))), \tag{1}$$

where $\mathcal{D}_{KL}$ is Kullback-Leibler (KL) divergence. $\mathcal{L}_{KL}$ gradually converges during the training process.

Since in the meta-training process, we employ a backbone with shared weights to extract features from the support and query samples, the above anatomical prior generation can be applied in parallel to the support and query branches. Similarly, the label encoder also adopts such a parallel manner to encode support and query labels and has the same structure as the feature encoder. It is worth noting that in the testing phase, the generation of latent factors only relies on the input support and query images. While during the training process, we exploit the input support and query images and the corresponding labels to learn the posterior distribution. As shown in Fig. 2, $\mathcal{L}_{KL}^s$ and $\mathcal{L}_{KL}^q$ represent the KL divergence on the support branch and query branch, respectively.

**Feature enrichment with anatomical prior.** To better guide the activation of anatomical targets, motivated by FiLM [20], we devise the feature enrichment module (FEM) with anatomical prior.

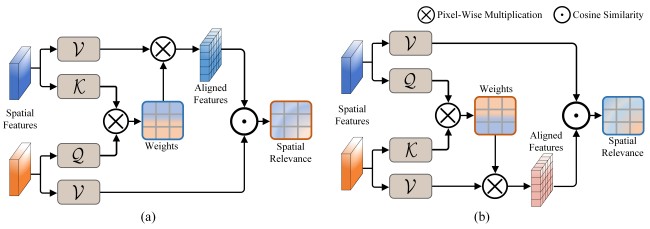

**Figure 3: (a) and (b) are two symmetric information interactions in the spatial interaction module (SIM) and are performed in parallel.**

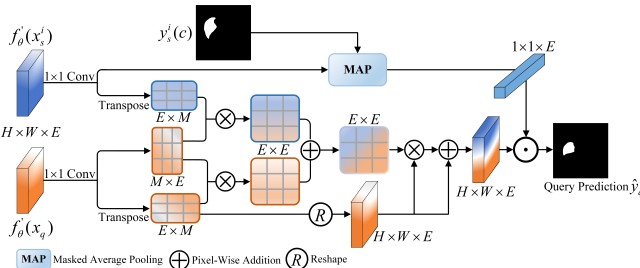

**Figure 4: The diagram of mutual guidance decoder (MGD).**

The module aims to perform the affine transformation on the embedded features according to the data sampled from the anatomical approximate posterior probability distribution to achieve feature enhancement. The rescaling and translation factors used for the affine transformation are predicted with the latent factor $z_x$. To be specific, $z_x$ is first sampled from latent embedding space, and then fed separately into two 3-layer MLPs with different parameter initializations to yield the scale $\beta$ and the shift $\alpha$. For convenience, $\beta$ and $\alpha$ are expanded to the same spatial resolution as the embedding feature $f(x)$. Afterwards, the affine-transformed features are passed to a convolutional layer to obtain enhanced features with anatomical prior, and this procedure is expressed as follows:

$$f'_t(x) = \beta f_t(x) + \alpha, \tag{2}$$

where the subscript $t$ denotes the feature map from the $t$-th channel in $f(x)$. Likewise, the above feature enrichment process based on anatomical prior can be applied to both support and query branches to produce new support and query features.

**Spatial interaction module.** In few-shot settings, segmentation models are prone to suffer from the spatial inconsistency problem between supports and query targets, and most FSS methods exploit multi-level spatial information to alleviate this problem. However, since multi-level spatial information cannot fundamentally affect the feature space, which tends to only achieve suboptimal solutions. For this purpose, we present a novel spatial contrastive learning strategy to obtain a feature space sensitive enough to spatial information, which optimizes the model via the spatial similarity between pairs of given support and query samples. To effectively activate spatial information, inspired by the Transformer [31], we design a spatial interaction module (Fig. 3), which can compare support and query features from the perspective of spatial information and construct a spatial contrastive loss. Taking Fig. 3 (a) as an example, assume $g_q$, $g_k$ and $g_v$ denote the query, key, and value encoders, which take the enhanced features $f'(x)$ as input and then output the query $Q$, key $\mathcal{K}$ and value $\mathcal{V}$, respectively:

$$Q = g_q(f'(x)), \mathcal{K} = g_k(f'(x)), \mathcal{V} = g_v(f'(x)), \tag{3}$$

where $g_q$, $g_k$ and $g_v$ are implemented as 1×1 convolutions, and the embedding dimension of $Q, \mathcal{K}, \mathcal{V} \in \mathbb{R}^{HW \times E'}$ is denoted by $E'$.

Given a support-query feature pair (*i.e.*, $f'(x_s)$ and $f'(x_q)$), SIM aims to obtain spatially aligned values of $f'(x_s)$ relative to $f'(x_q)$, regarded as $\mathcal{V}_{s|q}$. This spatial alignment process consists of two steps. It first obtains the spatial attention weight $W_{s,q} \in \mathbb{R}^{HW \times HW}$ by calculating the affinity matrix between the query $Q_s$ of $f'(x_s)$

and the key $\mathcal{K}_q$ of $f'(x_q)$). Afterwards, $W_{s,q}$ is employed to weight $\mathcal{V}_s$ to yield $\mathcal{V}_{s|q}$. The process can be mathematically defined as:

$$W_{s,q} = \text{softmax}(Q_s \mathcal{K}_q^\top / \sqrt{E'}), \tag{4}$$

$$\mathcal{V}_{s|q} = W_{s,q} \mathcal{V}_s, \tag{5}$$

Likewise, by swapping the roles of support and query samples, we can also obtain the alignment value $\mathcal{V}_{q|s}$ of $f'(x_q)$ relative to $f'(x_s)$.

**Spatial contrastive learning.** With the alignment values $\mathcal{V}_{s|q}$ and $\mathcal{V}_{q|s}$ for the support and query features, we first perform $l_2$ normalization on the value $\mathcal{V}^l$ of each spatial location $l$, where $\mathcal{V}^l$ indicates the feature vector with embedding dimension $E'$. Afterwards, the spatial relevance $r(f'(x_s), f'(x_q))$ between a support-query feature pair can be written as:

$$r(f'(x_s), f'(x_q)) =$$
$$\frac{1}{HW} \sum_{l=1}^{HW} \left( \frac{(\mathcal{V}_s^l)^\top \mathcal{V}_{q|s}^l}{\| \mathcal{V}_s^l \| \; \| \mathcal{V}_{q|s}^l \|} + \frac{(\mathcal{V}_q^l)^\top \mathcal{V}_{s|q}^l}{\| \mathcal{V}_q^l \| \; \| \mathcal{V}_{s|q}^l \|} \right), \tag{6}$$

Based on the spatial relevance $r$, we construct a novel spatial contrastive learning:

$$\mathcal{L}_{ij} = -\log \frac{\exp(r(f'(x_s^i), f'(x_q^j))/\tau)}{\sum\limits_{k=1}^{2N} \mathbb{I}_{i \neq k} \exp(r(f'(x_s^i), f'(x_q^k))/\tau)}, \tag{7}$$

$$\mathcal{L}_{SCL} = \frac{1}{2N-1} \sum_{i=1}^{2N} \sum_{j=1}^{2N} \mathbb{I}_{c_i = c_j} \mathcal{L}_{ij}, \tag{8}$$

where $\tau$ is the temperature coefficient and $\mathbb{I} \in \{0, 1\}$ is an indicative function.

### 3.3 Mutual Guidance Decoder

To obtain segmentation results, existing methods directly utilize support samples to guide the final segmentation of the query image. Nevertheless, such methods ignore the reference value of support images relative to query images. Therefore, we devise a mutual guidance decoder (MGD) to enhance query features via information interaction and yield more accurate segmentation results.

To be specific, for the features of input support and query $f'(x_s)$, $f'(x_q) \in \mathbb{R}^{H \times W \times E}$, we first transform them into two embedding spaces $h_s, h_q$, where $Z_s = h_s(f'(x_s))$, $Z_q = h_q(f'(x_q))$. And $h_s$ and $h_q$ are implemented as 1×1 convolutions. Then, these features are reshaped to $\hat{Z}_s, \hat{Z}_q \in \mathbb{R}^{M \times E_1}$. Multiple matrix multiplications are

performed to yield self-attention and cross-attention respectively, as depicted in Fig. 4. These two attentions are aggregated together via pixel-wise multiplication operations, which is formulated as:

$$\alpha = \hat{Z}_q^{\mathrm{T}}\hat{Z}_q + \hat{Z}_s^{\mathrm{T}}\hat{Z}_q, \tag{9}$$

where the superscript T refers to the matrix transpose operation and $\alpha$ denotes the aggregated attention along the channel dimension.

Afterwards, the aggregated attention is employed to reweight the query feature $\hat{Z}_q$, mathematically defined as:

$$\tilde{Z}_q = \alpha\hat{Z}_q + \hat{Z}_q, \tag{10}$$

where $\tilde{Z}_q$ denotes the enhanced query features and the co-occurrent features are effectively highlighted in the support and query branches. Following [41], we take a non-parametric way to calculate the class prototypes, namely, masked average pooling. Given the support features and corresponding masks, the prototype $\mathcal{P}_c$ of class $c$ is computed as follows:

$$\mathcal{P}_c = \frac{1}{K}\sum_{k=1}^{K}\frac{\sum_{x,y}Z_s^{(k,x,y)}y_s^{(k,x,y)}(c)}{\sum_{x,y}y_s^{(k,x,y)}(c)}, \tag{11}$$

where $(x, y)$ represents the coordinates in the spatial dimension and $K$ refers to the quantity of support samples belonging to class $c$. Accordingly, with the augmented query feature $\tilde{Z}_q$ and support prototypes $\mathcal{P} = \{p_c | c \in C\}$, we compute the cosine similarity between them to obtain the final query prediction mask $\hat{y}_q$:

$$\hat{y}_q = \mathrm{softmax}(\mathrm{cosine}(\tilde{Z}_q, \mathcal{P})), \tag{12}$$

where $\mathrm{softmax}(\cdot)$ denotes the normalized exponential function.

### 3.4 Training Objective

In the proposed framework, we employ cross-entropy $\mathcal{L}_{CE}$ to evaluate the segmentation loss of the query image. Kullback-Leibler divergence $\mathcal{L}_{KL}$ and spatial contrastive loss $\mathcal{L}_{SCL}$ are employed as auxiliary losses to facilitate the model to learn embedding features with sufficient discriminability. Therefore, the total objective of APSCL is expressed as:

$$\mathcal{L} = \mathcal{L}_{CE} + \lambda_{KL}(\mathcal{L}_{KL}^s + \mathcal{L}_{KL}^q) + \lambda_{SCL}\mathcal{L}_{SCL}. \tag{13}$$

where the $\lambda_{KL}$ and $\lambda_{SCL}$ are the trade-off weights, and are experimentally specified as 0.1 and 0.07, respectively.

## 4 EXPERIMENTS

### 4.1 Experimental Setting

**Datasets.** We examine the performance of our APSCL on three publicly available medical benchmark datasets, namely, CHAOS-T2 [9], MS-CMRSeg [42], and Synapse [12]: (1) CHAOS-T2 is derived from the 2019 ISBI Combined Healthy Abdominal Organ Segmentation Challenge, which consists of 20 3D abdominal MRI scans with a total of 623 axial 2D slices per scan. (2) MS-CMRSeg is from the 2019 MICCAI Multi-sequence Cardiac MRI Segmentation Challenge, which includes 35 3D cardiac MRI scans with a mean of 13 2D slices per scan. (3) Synapse stems from the 2015 MICCAI Multi-Atlas Abdomen Labeling Challenge, which is comprised of 30 3D organ CT scans with a total of 3779 axial 2D slices per scan.

To simulate the paucity of annotated samples in clinical settings, we carry out all experiments in the 1-way 1-shot setting. In the

CHAOS-T2 and Synapse scans, we segment four classes: liver, left kidney (LK), right kidney (RK), and spleen. In the MS-CMRSeg scans, we segment three organ classes: left-ventricle blood pool (LV-BP), right-ventricle (RV), and left-ventricle myocardium (LV-MYO). We choose one of the organ classes as the unseen testing class (*i.e.*, *novel* class) and the others as the seen training classes (*i.e.*, *base* classes) to form multiple few-shot tasks. In all experiments, the five-fold cross-validation is employed to evaluate the models.

**Implementation details.** The proposed APSCL is implemented by PyTorch on an NVIDIA Tesla V100 GPU with 32 GB memory. A popular ResNet-101 [5] backbone is employed as the encoder $f(\cdot)$, which is pre-trained on the MS-COCO dataset [27] following the common practice [19]. It processes $3 \times 256 \times 256$ dimensional input data and generates $256 \times 32 \times 32$ dimensional embedding features. During anatomical prior generation, the number of latent factors is experimentally specified as 32. During training, the entire learning process is optimized by the stochastic gradient descent optimizer with a learning rate of $1e\text{-}3$ and a batch size of 1 in an end-to-end manner. The weight decay and momentum are specified as $5e\text{-}4$ and 0.9, respectively. The whole network is trained for 100 epochs, each containing 1000 episodes. The Dice score is adopted as the performance metric in the experiments.

### 4.2 Comparison with the Other Methods

In this section, we compare the proposed anatomical prior guided spatial contrastive learning (APSCL) and state-of-the-art FSS methods, including the baseline method SE-Net [23], PANet [33], SSL-ALPNet [19], PoissonSeg [25], RP-Net [29], GCN-DE [28], SRNet [34], AAS-DCL [35], ADNet [4], and LVQM [6]. All the models are reimplemented in the same setting for a fair comparison. Table 1 reports the 1-shot results with respect to the Dice score. It is observed that the proposed APSCL achieves excellent performance in the 1-shot scenario, exceeding the previous FSS method by a significant margin. For example, compared to the baseline method SE-Net, our APSCL achieves 35.98%, 42.95%, and 40.30% improvement in terms of the mean Dice score on CHAOS-T2, MS-CMRSeg, and Synapse, respectively. In comparison with the state-of-the-art method LVQM, our APSCL surpasses LVQM by an average of 3.61%, 2.30%, and 6.38% on CHAOS-T2, MS-CMRSeg, and Synapse, respectively. It is because our method fully exploits the anatomical information in medical slices to guide the model to learn discriminative features suitable for clinical practice via contrastive learning, and more efficiently exploits the information of support features to guide segmentation via mutual guidance decoder. These results on three datasets demonstrate that considering anatomical information in medical images and guidance information from support samples is crucial for improving medical image segmentation models in few-shot scenarios.

To intuitively illustrate the superiority of the proposed APSCL, we also present some examples of visual segmentation results of our APSCL and other methods on the CHAOS-T2 and Synapse datasets in Fig. 5. It can be observed that our proposed APSCL yields desirable segmentation masks for organs with various intensities, scales, and morphologies in complex backgrounds while other methods yield inferior segmentation results on most organs. Since the comparison methods do not take into account anatomical

**Table 1: Quantitative comparison (in Dice score %) of our APSCL and other methods in the 1-way 1-shot setting on CHAOS-T2, MS-CMRSeg, and Synapse datasets. The number in bold indicates the best segmentation result.**

| Method | CHAOS-T2 | | | | | MS-CMRSeg | | | | Synapse | | | | |
|---|---|---|---|---|---|---|---|---|---|---|---|---|---|---|
| | Liver | LK | RK | Spleen | Mean | RV | LV-BP | LV-MYO | Mean | Liver | LK | RK | Spleen | Mean |
| SE-Net [23] | 28.68 | 58.95 | 60.25 | 50.06 | 49.49 | 18.97 | 59.61 | 26.55 | 35.04 | 47.05 | 41.83 | 35.02 | 40.91 | 41.20 |
| PANet [33] | 47.39 | 54.29 | 42.68 | 50.42 | 48.70 | 57.62 | 70.24 | 43.91 | 57.26 | 40.27 | 33.22 | 19.61 | 31.78 | 31.22 |
| SSL-ALPNet [19] | 71.01 | 71.94 | 77.98 | 63.38 | 71.08 | 74.35 | 84.06 | 61.64 | 73.35 | 74.68 | 62.02 | 51.38 | 65.77 | 63.46 |
| PoissonSeg [25] | 60.06 | 53.98 | 59.63 | 56.83 | 57.63 | 68.41 | 79.82 | 51.56 | 66.60 | 56.08 | 52.83 | 49.40 | 53.37 | 52.92 |
| RP-Net [29] | 67.04 | 77.39 | 84.51 | 74.83 | 75.94 | 75.65 | 83.70 | 62.83 | 74.06 | 80.67 | 70.27 | 72.82 | 69.56 | 73.33 |
| GCN-DE [28] | 53.08 | 75.05 | 83.54 | 65.48 | 69.29 | 57.82 | 82.38 | 61.46 | 67.22 | 47.02 | 69.38 | 73.48 | 56.70 | 61.65 |
| SRNet [34] | 76.04 | 73.70 | 82.45 | 70.26 | 75.61 | 70.57 | 85.79 | 64.13 | 73.50 | 73.93 | 66.52 | 59.71 | 61.36 | 65.38 |
| AAS-DCL [35] | 72.78 | 52.58 | 83.38 | 60.93 | 67.42 | 76.11 | 84.77 | 63.27 | 74.72 | 72.40 | 63.80 | 68.04 | 67.01 | 67.81 |
| ADNet [4] | 80.69 | 78.31 | 87.31 | 75.85 | 80.54 | 70.57 | 83.16 | 57.18 | 70.30 | 75.80 | 68.26 | 64.70 | 60.74 | 67.38 |
| LVQM [6] | 83.08 | 80.01 | 87.54 | 76.79 | 81.86 | 76.23 | 87.15 | 63.68 | 75.69 | 80.43 | 73.14 | 76.10 | 70.81 | 75.12 |
| APSCL (Ours) | **86.73** | **84.66** | **89.66** | **80.82** | **85.47** | **77.49** | **90.79** | **65.68** | **77.99** | **87.74** | **80.19** | **78.00** | **80.05** | **81.50** |

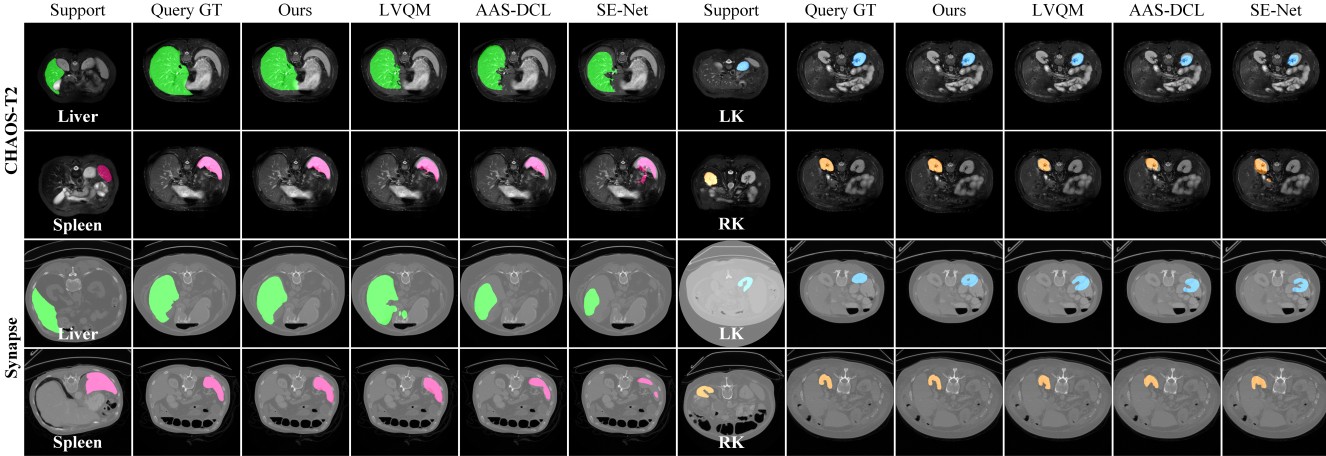

**Figure 5: Qualitative comparison of our APSCL and other methods in the 1-way 1-shot setting on the CHAOS-T2 and Synapse datasets. The segmentation results of our APSCL are closer to the ground-truth than those of other methods. GT indicates the ground truth.**

prior information from medical images, segmentation masks are typically incomplete and inaccurate. This further proves the effectiveness of our APSCL for medical image segmentation in low-data regimes.

## 4.3  Ablation Study

In this section, we present ablation study results under the 1-way 1-shot scenario on the CHAOS-T2 dataset.

**Effect of major components.** As illustrated in Table 2, we investigate the impact of the major components by incrementally applying our proposed components to the baseline model, *i.e.*, SE-Net. Firstly, integrating each component individually into the baseline model can achieve performance gains. In particular, the FEM and SCL can improve the mean Dice score by 12.67% and 17.16%, respectively. When applying the two components together to the baseline model, it is observed that the FEM & SCL can substantially enhance the mean Dice score to 78.93%, while the FEM & MGD and SCL & MGD slightly boost segmentation performance. Moreover,

**Table 2: Ablative results (in Dice score %) of various components of the proposed method on the CHAOS-T2 dataset. The baseline method is SE-Net [23], namely the APSCL without the FEM, SCL, and MGD modules.**

| FEM | SCL | MGD | Liver | LK | RK | Spleen | Mean |
|---|---|---|---|---|---|---|---|
| | | | 28.68 | 58.95 | 60.25 | 50.06 | 49.49 |
| ✓ | | | 64.74 | 63.48 | 58.78 | 61.65 | 62.16 |
| | ✓ | | 68.90 | 66.87 | 64.39 | 66.45 | 66.65 |
| | | ✓ | 61.92 | 57.79 | 62.74 | 59.95 | 60.60 |
| ✓ | ✓ | | 81.79 | 76.09 | 84.72 | 73.10 | 78.93 |
| | ✓ | ✓ | 79.55 | 71.87 | 76.01 | 68.72 | 74.04 |
| ✓ | | ✓ | 74.60 | 73.78 | 75.90 | 61.65 | 71.48 |
| ✓ | ✓ | ✓ | **86.73** | **84.66** | **89.66** | **80.82** | **85.47** |

the mean Dice score is further boosted to 85.47% when the above three parts are jointly integrated into the baseline model. The above

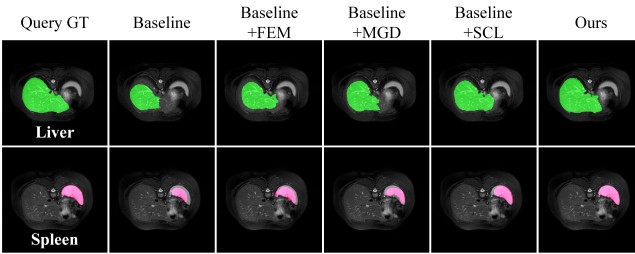

**Figure 6: Visual comparison of the role of the various modules on the CHAOS-T2 dataset. Compared with the baseline, the proposed APSCL takes into account additional anatomical prior from the ground-truth labels and yields more accurate segmentation results.**

**Table 3: The effect of $\lambda_{KL}$ and $\lambda_{SCL}$ on the performance of the proposed method.**

| $\lambda_{KL}$ | $\lambda_{SCL}$ | Liver | LK | RK | Spleen | Mean |
|---|---|---|---|---|---|---|
| 0.08 | | 86.30 | 84.09 | 88.19 | 79.13 | 84.43 |
| 0.1 | 0.07 | **86.73** | **84.66** | **89.66** | **80.82** | **85.47** |
| 0.12 | | 85.23 | 83.12 | 87.45 | 79.80 | 83.90 |
| | 0.05 | 85.52 | 83.17 | 88.71 | 79.33 | 84.18 |
| 0.1 | 0.07 | **86.73** | **84.66** | **89.66** | **80.82** | **85.47** |
| | 0.09 | 86.53 | 83.57 | 89.31 | 80.15 | 84.89 |

observations validate the interactive superiorities of the FEM, SCL, and MGD.

In addition, we also present visualization results of ablation experiments, as shown in Fig. 6. It can be intuitively observed that, in comparison to the baseline method, the anatomical prior generated by distribution consistency can lead to better segmentation masks with the assistance of the FEM. And after integrating the three essential components together into the baseline method, the segmentation masks are closer to the ground-truth masks. These visualization results further verify the effectiveness of the proposed three components.

**Visualization of pixel features.** We project the obtained pixel features into 2D space using t-SNE [30]. As shown in Fig. 7, the learned pixel features by APSCL become more compact and well separated compared to the baseline (*i.e.*, SE-Net). This proves that the proposed APSCL can yield better segmentation performance for the unseen classes.

**Impact of the hyper-parameters $\lambda_{KL}$ and $\lambda_{SCL}$.** In this experiment, we examine the impact of the trade-off hyper-parameters $\lambda_{KL}$ and $\lambda_{SCL}$ in the total objective function, which control the effect of KL divergence and spatial contrastive loss in Eq. (13). The experiments are conducted on the CHAOS-T2 dataset in the 1-way 1-shot setting. Table 3 presents the performance of the different hyper-parameter values of $\lambda_{KL}$ and $\lambda_{SCL}$ respectively. As the hyper-parameter values increase, $\lambda_{KL}$ and and $\lambda_{SCL}$ exhibit favorable and then unfavorable effect, while both reasonable values exhibit a better upper bound on performance. Our APSCL is robust enough to the selection of hyper-parameters, and it reaches the optimal

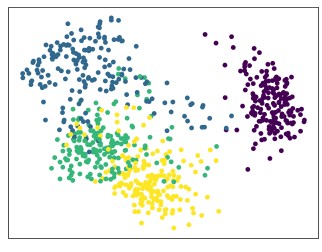
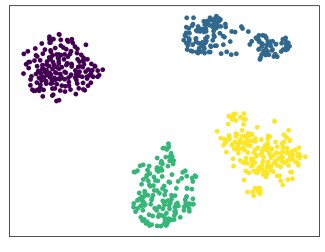

(a) Baseline     (b) Our Method

**Figure 7: The t-SNE [30] visualization on CHAOS-T2 dataset of the pixel features learned by various methods. Colors denote pixel classes.**

**Table 4: Performance analysis of the number of latent factors on the CHAOS-T2 dataset.**

| Latent Factors | FLOPs | Mean Dice Score | Inference Time | Training Time |
|---|---|---|---|---|
| 2 | 34.62G | 73.74 | 27.1ms | 12.31h |
| 4 | 34.98G | 75.48 | 31.3ms | 12.43h |
| 8 | 36.01G | 78.46 | 43.6ms | 12.83h |
| 16 | 38.12G | 82.96 | 65.3ms | 13.69h |
| 32 | 41.78G | 85.47 | 92.8ms | 14.96h |
| 64 | 46.25G | 85.84 | 151.9ms | 16.51h |
| 128 | 53.13G | 85.90 | 241.3ms | 17.88h |

performance when $\lambda_{KL}$ = 0.1 and $\lambda_{SCL}$ = 0.07. For convenience, the values of $\lambda_{KL}$ and $\lambda_{SCL}$ are fixed in all experiments.

**Influence of the size of the latent factor $z_x$.** We also investigated the influence of the size of the latent factor $z_x$, and the experimental results are illustrated in Table 4. The latent factor is used for anatomical prior generation. The ablation experiments are carried out in the 1-way 1-shot case on the CHAOS-T2 dataset. From Table. 4, it is observed that as the size of the latent factor is close to 32, the growth of the mean Dice score slows down while the growth in computational complexity remains large. Considering computation efficiency and performance, we set the size of the latent factor $z_x$ to 32 in all experiments.

## 5 CONCLUSION

In this work, we propose a novel anatomical prior guided spatial contrastive learning scheme, named APSCL, to tackle the challenging medical image segmentation tasks in extremely low-data regimes. Unlike existing FSS methods, our framework fully exploits anatomical prior information from medical images to construct contrastive learning from a spatial perspective to boost the distinguishability of the learned features. Moreover, the mutual guidance decoder is designed to fully exploit the guidance information of support samples to yield more accurate segmentation results. Our extensive experiments on CHAOS-T2, MS-CMRSeg and Synapse datasets demonstrate that the proposed APSCL surpasses current state-of-the-art FSS methods by as much as 6.38% with respect to the Dice score.

# 6 ACKNOWLEDGMENTS

This work was partially supported by the National Natural Science Foundation of China under Grant No. 62172067, the National Key Research and Development Project under Grant No. 2019YFE0110800, the Natural Science Foundation of Chongqing for Distinguished Young Scholars under Grant No. CSTB2022NSCQ-JQX0001, and the Chongqing University of Posts and Telecommunications Ph.D. Innovative Talents Project under Grant No. BYJS202308.

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
