# OpenReview forum: "Anatomical Prior Guided Spatial Contrastive Learning for Few-Shot Medical Image Segmentation"
_acmmm.org/ACMMM/2024/Conference — MM2024 Poster_

### Official Review · Reviewer_mCF6 · 2024-05-20

**Rating:** 4
**Confidence:** 2

**Summary:**

Few-shot semantic segmentation has considerable potential for lowdata scenarios, especially for medical images that require expertlevel dense annotations. Existing few-shot medical image segmentation methods strive to deal with the task by means of prototype learning, but ignored the rich anatomical prior knowledge in medical images, which hinders effective feature enhancement for medical images. In this paper, authors propose an anatomical prior guided spatial contrastive learning, called APSCL, which exploits anatomical prior knowledge derived from medical images to construct contrastive learning from a spatial perspective for few-shot medical image segmentation. Comprehensive experiments on three challenging medical image datasets, CHAOS-T2, MS-CMRSeg, and Synapse, prove that APSCL significantly surpasses state-of-the-art few-shot medical segmentation methods, with a mean improvement of 3.61%, 2.30%, and 6.38% on the Dice score, respectively.

**Strengths:**

1.	This paper proposes to use the anatomical prior knowledge to guide spatial contrastive learning to help achieve better medical image segmentation results in FSS.
2.	The validation experiments  are extensive and enough.
3.	The paper content is rich in graphics and tables, with clear formulas listed,

**Limitations:**

1. Is the dataset used in the experiment consistent with the dataset used in LVQM[1]? The first two datasets seem to be the same dataset according to the description?
2. If the experimental dataset is consistent, why are the experimental results inconsistent in this paper and LVQM[1]?
3. The visualization results of the experiment in Figure 5 would be better if supplemented with MS-CMRSeg.
4. A more detailed explanation is needed for the calculation of LKL, such as how the Prior Block and Posterior Block used are implemented? No explanation found in the text.

[1] Shiqi Huang, Tingfa Xu, Ning Shen, Feng Mu, and Jianan Li.  Rethinking Few-Shot Medical Segmentation: A Vector Quantization View. In Proceedings of the IEEE/CVF Conference on Computer Vision and Pattern Recognition,2023, 3072–3081.

**Suitability:**

2

---

### Official Review · Reviewer_48LM · 2024-05-25

**Rating:** 5
**Confidence:** 3

**Summary:**

This paper tackles the challenge of accurately segmenting medical images when limited training data is available. It introduces an innovative method called Anatomical Prior guided Spatial Contrastive Learning (APSCL) that leverages anatomical knowledge and spatial relationships to boost segmentation performance in low-data scenarios.

**Strengths:**

APSCL is a novel framework incorporating anatomical priors and spatial contrastive learning, representing a great contribution to medical image analysis.

Compared to state-of-the-art methods, APSCL demonstrates significant improvements in segmentation accuracy across three challenging datasets: CHAOS-T2 (3.61% higher Dice score), MS-CMRSeg (2.30% higher), and Synapse (6.38% higher).


The end-to-end training approach simplifies the implementation process, potentially making APSCL more adaptable to various datasets and use cases.

**Limitations:**

Integrating anatomical priors and spatial contrastive learning may make APSCL complex to implement. Specifically, there are three major components: FEM, SCL, and MGD. Some of these designs probably could be simplified. For example, the ablation study in Table 2 shows that MGD has the weakest effect. How would it be different with simple pooling and some linear layer, instead of mutual attention and gated pooling?


For effective SCL, the method's effectiveness heavily depends on the quality and accuracy of the available anatomical prior knowledge. Poor-quality priors could diminish performance gains. The paper uses an effective kl regularization term for spatial contrastive learning and few-shot segmentation. However, this consistency is conducted in low-dimensional space after feature compression. The potential weakness could be the loss of fine-grained morphology prior in such space.

Model comparison: Table 1 shows the proposed method, which achieved remarkable results over the baseline methods. The latest work in Table 1 is from CVPR 2023, which is insufficient to represent the up-to-date progress. Some recent works have not been compared or at least acknowledged. For example:
Tianang Leng, Yiming Zhang, Kun Han, Xiaohui Xie; Self-Sampling Meta SAM: Enhancing Few-Shot Medical Image Segmentation With Meta-Learning. Proceedings of the IEEE/CVF Winter Conference on Applications of Computer Vision (WACV), 2024, pp. 7925-7935


It would be interesting to see the potential limitations in the conclusion and to inspire future work.

**Suitability:**

3

---

### Official Review · Reviewer_bgkH · 2024-05-27

**Rating:** 4
**Confidence:** 3

**Summary:**

The paper titled "Anatomical Prior Guided Spatial Contrastive Learning for Few-Shot Medical Image Segmentation" introduces a novel approach to improve few-shot medical image segmentation, which iscalled APSCL (Anatomical Prior Guided Spatial Contrastive Learning). It leverages anatomical prior knowledge to guide spatial contrastive learning. This method enhances feature discriminability by aligning features with embedded anatomical representations. Additionally, the framework includes a mutual guidance decoder that utilizes support sample information to predict pixel labels in query images effectively. The APSCL model can be trained end-to-end using episodic training and demonstrates significant performance improvements over existing methods on three challenging medical image datasets: CHAOS-T2, MS-CMRSeg, and Synapse.

**Strengths:**

- **Innovative Approach**: The use of anatomical prior knowledge and spatial contrastive learning is novel for the specific challenges of medical image segmentation in low-data scenarios.
- **Comprehensive Experiments**: The method was tested on three challenging medical image datasets, showing substantial improvements over state-of-the-art methods, which validates the robustness and generalizability of the proposed framework.
- **Detailed Ablation Studies**: The paper includes thorough ablation studies that highlight the contributions of different components of the APSCL framework, strengthening the validity of their design choices.
- **Presentation**: Well-organized presentation.

**Limitations:**

1. Spatial Contrastive Learning: From line 455-457, the spatial inconsistency problem is suggested. What is a concrete example of it? For the spatial features Q,K,V, W, what are they represented for? I am not sure about the reasons behind using attention mechanism and asking two spatial features to be the same.
2. Anatomical Posterior: How do you model the latent distribution? I would assume the latent variable is just 1-D vectors, though in figure2, it is illustrated as a 2D map (line 352). Also, How is the use of latent method related to the term "anatomy"?
3. Ablation study: Is there an ablation experiment for the use of anatomical prior? Replacing the current latent method on spatial prior and anatomical posterior with other segmentation loss to find out if it is the anatomic factor,  rather than the network structure, that works here.

**Suitability:**

3

---

### Official Review · Reviewer_5Fyw · 2024-05-28

**Rating:** 5
**Confidence:** 3

**Summary:**

The paper introduces a novel algorithm known as Anatomical Prior Guided Spatial Contrastive Learning (APSCL), which leverages anatomical prior knowledge to enhance feature representation for few-shot medical image segmentation. The APSCL algorithm uniquely incorporates anatomical priors by enforcing distribution consistency, allowing the model to perform affine transformations on embedded features based on data sampled from the anatomical approximate posterior probability distribution. This process significantly enhances feature alignment and representation. Additionally, the paper proposes the use of a Spatial Interaction Module that utilizes a transformer mechanism to facilitate long-range interactions between support and query images, enhancing the spatial contrastive learning process before moving into a mutual guidance decoder for segmentation.

**Strengths:**

Novelty:
1. Innovative Use of Anatomical Priors: The APSCL framework is pioneering in its integration of anatomical priors with spatial contrastive learning, making it a significant departure from traditional prototype-based approaches in few-shot segmentation. This integration allows for an advanced feature alignment and enhancement that is specifically tailored to the complex requirements of medical imaging.
2. Spatial Interaction via Transformers: The incorporation of a transformer-based Spatial Interaction Module not only facilitates the generation of interaction features but also crucially models significant spatial deformations and misalignments that are not adequately addressed by the prior-driven affine transformations. This capability significantly enhances the model's ability to handle the dynamic and varied nature of medical image segmentation, where traditional methods may fall short.

Experimental Performance:
1. Superior Performance Metrics: The method demonstrates a substantial improvement over existing state-of-the-art approaches on three challenging medical image datasets—achieving mean improvements of 3.61%, 2.30%, and 6.38% on the Dice score respectively. These results underscore the effectiveness of the proposed method in practical, real-world scenarios.
2. Comprehensive Validation: Through rigorous experiments, including ablation studies(Table 2), the paper effectively demonstrates the critical components and overall efficacy of the APSCL framework. Furthermore, the exploration of weakly-supervised learning scenarios adds another layer of robustness to the validation, showing the versatility and adaptability of the approach.

**Limitations:**

1. Detailed Explanation of Anatomical Priors Generation: The generation of anatomical priors, particularly the optimization of equation (1), requires more detailed clarification. It would be beneficial to specify the parametric form of the posterior probability distributions used, whether they are modeled using Gaussians or other probabilistic forms. For enhanced clarity, especially for readers unfamiliar with this methodology, a more thorough explanation in the main text or an appendix would be advisable.

2. Enhanced Visualization: The paper would greatly benefit from the inclusion of visualizations for the anatomical priors, the resulting affine transformations, and the attention mechanisms within the Spatial Interaction Module and Mutual Guidance Decoder. Given that these elements are central to the paper’s contributions and distinguish it from prior work on anatomical prior-based few-shot segmentation, detailed visual representations could significantly enhance understanding and appreciation of the novel approaches.

3. Inclusion of Experimental Baselines and Failure Case Analysis: The absence of experimental comparisons with fully supervised segmentation models and an analysis of failure cases is a notable limitation. Understanding the performance ceiling with fully supervised approaches on the same datasets could provide valuable context for the few-shot results, highlighting the performance gap and potential areas for improvement. Additionally, an analysis of situations where the proposed method excels or falls short, particularly across different imaging types, organs, or modalities, would offer deeper insights into the method’s limitations and inform future research directions.

4. Comprehensive Literature Review: The paper's review of related literature appears incomplete, particularly concerning recent advancements in anatomical-prior-driven few-shot and medical image segmentation. Works such as "Learning what and where to segment: A new perspective on medical image few-shot segmentation, Medical Image Analysis 2023" and "A location-sensitive local prototype network for few-shot medical image segmentation, ISBI 2021", as well as other work on anatomical-prior-based techniques could be reviewed and discussed.

**Suitability:**

2

---

### Meta-Review · Area_Chair_ck86 · 2024-06-29

**Recommendation:** Accept (Poster)
**Confidence:** 4

**Metareview:**

The paper presents an innovative approach called Anatomical Prior Guided Spatial Contrastive Learning (APSCL) for few-shot medical image segmentation. Reviewers recognized the novelty and effectiveness of integrating anatomical prior knowledge with spatial contrastive learning, resulting in substantial performance improvements across multiple datasets. To further enhance the paper, reviewers suggested providing more detailed explanations of the anatomical priors generation, improving visualizations of key components, comparing with additional recent works, and including failure case analysis. The authors addressed these concerns in the rebuttal, leading to a consensus for acceptance, with the recommendation to incorporate these suggested improvements for increased clarity and impact.